# Exploring Multi-Target Therapeutic Strategies for Glioblastoma via Endogenous Network Modeling

**DOI:** 10.3390/ijms26073283

**Published:** 2025-04-01

**Authors:** Mengchao Yao, Xiaomei Zhu, Yong-Cong Chen, Guo-Hong Yang, Ping Ao

**Affiliations:** 1Shanghai Center for Quantitative Life Sciences and Physics Department, Shanghai University, Shanghai 200444, China; eric_yao@shu.edu.cn; 2Shanghai Key Laboratory of Modern Optical System, School of Optical-Electrical and Computer Engineering, University of Shanghai for Science and Technology, Shanghai 200444, China; 3Department of Physics, Shanghai University, Shanghai 200444, China; ghyang@shu.edu.cn; 4College of Biomedical Engineering, Sichuan University, Chengdu 610065, China

**Keywords:** glioma, multi-target therapy, endogenous network theory, landscape, system biology

## Abstract

Medical treatment of glioblastoma presents a significant challenge. A conventional medication has limited effectiveness, and a single-target therapy is usually effective only in the early stage of the treatment. Recently, there has been increasing focus on multi-target therapies, but the vast range of possible combinations makes clinical experimentation and implementation difficult. From the perspective of systems biology, this study conducted simulations for multi-target glioblastoma therapy based on dynamic analysis of previously established endogenous networks, validated with glioblastoma single-cell RNA sequencing data. Several potentially effective target combinations were identified. The findings also highlight the necessity of multi-target rather than single-target intervention strategies in cancer treatment, as well as the promise in clinical applications and personalized therapies.

## 1. Introduction

Glioblastoma multiforme (GBM) is the most common primary tumor of the central nervous system [1,2,3], characterized by poor prognosis, short survival time, and high resistance to treatment [4,5]. Current options for GBM treatment are rather restricted, with the most commonly used agent, temozolomide (TMZ), offering only limited efficacy, yet with significant side effects [5,6,7,8,9]. Targeted therapies, including RTK inhibitors, PI3K/Akt pathway inhibitors, mTOR pathway inhibitors, and CDK4/6 inhibitors, are being explored as alternatives [5,10,11].

Despite their initial effectiveness, single-target therapies often result in resistance over time [6,12]. Recent studies highlight that tumorigenesis is not driven predominantly by some single-gene mutations or isolated molecular mechanisms; instead, it is associated with multi-factorial and multilevel biological processes. Tumor cells exhibit high adaptability and heterogeneity due to interactions among genetic, epigenetic, metabolic, and micro-environmental factors [13]. As such, conventional linear approaches and single-target strategies fail to fully comprehend the complexity of tumor biology. Therefore, researchers have shifted their focus toward exploring multi-target therapeutic strategies for cancer [14,15,16,17], including but not limited to the augmentation of immunotherapeutic interventions [18] and the induction of tumor differentiation [19].

Treating cancer from a systemic perspective has received increasing attention in recent years [20]. Analyses based on systems biology may offer a perspective distinct from the linear and reductionist approach to understanding tumor complexity by recognizing that the numerous components within a biological organism form complex and intersecting networks [21]. The interplay of these networks can exhibit characteristics such as self-organization, nonlinearity, and dynamic evolution. Endogenous network theory (ENT), a novel approach based on such concepts, emphasizes the causality of complexity and posits the existence of a core endogenous network whose nonlinear dynamics may be characterized via “energy potential”, serving as a quantitative representation of Waddington’s epigenetic landscape [22,23]. This theory has been applied to a number of cases of tumors [23,24] and developmental progress [25,26].

The landscape of glioma has been established based on experimental data-driven [21] and first-principle-like analyses [27], thereby advancing the understanding of the mechanisms underlying glioma initiation and progression. Nevertheless, leveraging it to inform therapeutic strategies remains elusive. Multi-target therapies may be clinically promising, but would involve exploring a vast number of combinations, making the task challenging to complete by experiment alone [6,12,17,28]. Efforts have been made to explore tumor multi-target therapies based on correlation networks [29], yet the fundamental gap between correlation and causation demands a causality-driven framework to bridge it. Although the potential of network dynamics-based approaches for tumor multi-target therapy has long been discussed [22,30], a concrete implementation strategy remains lacking. This situation calls for innovative methodologies to search for valid strategies for multi-target cancer therapies for glioma.

A recent study demonstrated the utility of causal tumor intervention based on the endogenous network approach in identifying potential multi-target strategies for gastric cancer [31]. In this article, we systematically explored multi-target therapeutic strategies for glioblastoma using such causal-dynamic methodology, aiming to provide an analytical framework for effective and precise treatment of glioma.

This article is organized into several parts as follows: Introduction, Results, Model Construction and Dynamics, Model Validation, Model Predictions (Exploration of Therapeutic Strategies), Discussion, Materials and Methods, and Conclusions. In Section 2.2, we use published data [32] and prior knowledge [33] to validate the model results under the new regulation of η. In Section 2.3, we explore the combination space of multi-target therapy strategies and identify several potential intervention combinations for glioma treatment.

## 2. Results

### 2.1. Model Construction and Dynamics

An endogenous network for glioma was constructed based on prior experimental knowledge (Figure 1A) and subsequent validation at multiple levels [27]. Several stable states in this model—solutions that spontaneously return to their original state after small perturbations—correspond to different types of glioma cells and apoptotic glial cells, respectively. Concentrations/activities of the network nodes were regulated by Hill functions, with the degradation rates set to 1 in the previous work [27]. In the current work, we further introduced dynamics on the degradation times with a random initial distribution to simulate the regulatory process in the biological system (Figure 1B). Interventions and mutations were simulated by setting the regulation function of the related nodes to extreme values: 0 for inhibition or mutation, and 1 for activation.

### 2.2. Model Validation

#### 2.2.1. Validation with scRNA-Seq Data

In the absence of intervention (i.e., treatment or mutation), our model identified four clusters of stable states (Figure 2A,C). For these results obtained from our current calculations, hierarchical clustering combined with biological knowledge suggests that the main distinctions are between proliferation/apoptosis and astrocytic-like/oligodendrocytic-like states (Figure 2A). Principal component analysis (PCA) is sufficient to distinguish these two main features (Figure 2C). Two of them closely matched a set of published scRNA-seq results of glioblastoma [32] (Figure 2A–C), as tumor states (PC1 < 1), constituting ~28.32% of all solutions (Table 1). The remaining two exhibited high apoptotic signals, suggesting that they correspond to apoptotic states.

#### 2.2.2. Common Mutations in Glioblastoma Expand Tumor Stable States

As a part of the multi-level validation, we then simulated the impact of common glioblastoma mutations [33] on the distribution of the stable states, including inactivation of PTEN, P27/P21, P53, CDKN2A, and RB, as well as over-expression of Akt, Ras, CDK4, and HIF. These interventions all significantly expanded the proportion of tumor stable states (Figure 2G–I, Table 1). Such consistency with clinical/experimental observations further consolidated the model constructed for the subsequent investigation.

### 2.3. Exploration of Therapeutic Strategies

#### 2.3.1. Single-Target Interventions

After the validation, we next investigated potential multi-target therapeutic strategies. First, we went through all single-target interventions and found that the downregulation of Akt achieved the best therapeutic effect, reducing tumor states from 28.32% to 2.29% (Figure 2D, Table 1). Other interventions, such as upregulation of CDKN2A (4.06%) (Figure 2E, Table 1), P53 (6.09%), and PTEN (7.56%), also significantly reduced tumor state proportions (Table 1). However, no single-target intervention could entirely eliminate tumor stable states. These results will be further discussed later.

#### 2.3.2. Multi-Target Interventions

We then explored 280 random dual-target combinations. Notably, combining Akt downregulation with P53 upregulation completely eliminated tumor stable states (Figure 2F, Table 2). Additionally, 15 dual-target combinations yielded better results than the best output in single-target simulation (Table 2).

Considering that aberrations in Akt and P53-related signaling pathways are common in glioblastoma [33], which could result in the decoupling of these interventions from downstream regulation, we further investigated triple-target intervention combinations. In all 500 random combinations we explored, 8 triple-target interventions completely eliminated tumor stable states (Figure 2J–Q, Table 2), and 29 combinations outperformed the best output in single-target simulation (partially shown in Table 2; full results can be found in Appendix A).

Additionally, we identified certain intervention combinations that significantly increased the proportion of tumor stable states. In the dual-target intervention simulations, upregulating Akt while downregulating CDKN2A resulted in the tumor stable state proportion expanding to 84.73% (Figure 2I). Among the total of 280 dual-target simulations, 145 combinations (51.79%) increased the tumor stable state proportion, with 12 combinations (4.29%) bringing it above 50% (Partially shown in Table 2). In the simulations involving 500 triple-target combinations, 326 interventions (65.2%) expanded the tumor stable state proportion, with 182 interventions (36.4%) pushing it over 50%, 33 combinations (6.6%) that completely eliminated non-tumor stable states (partially shown in Table 2; full results can be found in Appendix A).

## 3. Discussion

### 3.1. Model Limitation

While the model demonstrates strong predictive capabilities, several limitations remain. The coarse-grained model described in this work was a simplification of the underlying biological system with a focus on key pathways and modules [22,23]. The reduction might overlook important regulatory factors or feedback mechanisms, and could prevent certain critical targets outside the model from being detected. This issue will need to be addressed in future work by expanding and refining the model step by step. In addition, experimental knowledge is insufficient to truly determine the edge weights, which are detailed parameters. Instead of fitting data, we started from a theoretical basis, assuming that the nodes and edges within the core network are “equally important”. Therefore, after normalizing the concentration/activity of the nodes, the weights of all edges are considered to be the same [22,23]. Though the stochastic approach partially addresses the biological variability, the dynamics might significantly deviate from the behavior of real-world biology. These limitations highlight the need for collaboration between theoretical and experimental approaches. To a fair extent, the modeling results did align with a set of published scRNA-seq data and existing knowledge. More theory–experiment collaboration may lead to a more precise scope of the combination targets.

We chose a linear method, principal component analysis [34], rather than the more often used nonlinear dimensionality reduction methods like t-SNE or UMAP, to compare the theoretical results with single-cell data. For the steady states obtained from our current calculations, hierarchical clustering combined with biological knowledge suggests that the main distinctions are between proliferation/apoptosis and astrocytic-like/oligodendrocytic-like states, and PCA is sufficient to distinguish these two main features. The interpretable and reproducible nature of PCA [34] also makes it more suitable for observing how different interventions impact the simulated results under the same standard.

Yes, the dimensionality reduction results from PCA are far less detailed than those from UMAP/t-SNE. Therefore, from a data-driven perspective, using only PCA is likely to overlook characteristics such as tumor heterogeneity and cell state transitions. However, in the context of this study, our method is not limited to data analysis. We adopt a bottom-up perspective, where experimental data analysis serves to validate theoretical predictions. Furthermore, in this study, we are more concerned with commonalities than with subtle differences. Given the orders-of-magnitude disparity between the number of coarse-grained nodes in our model [25] and the number of functional genes in biological systems (tens of thousands), as well as the limited experimental knowledge, discrepancies between the theory output and experimental observation are inevitable. Nevertheless, the consistency between the theory and experimental results—both qualitatively (e.g., the alignment of common glioma variations with tumor attractor landscape trends) and quantitatively (e.g., the overlap between tumor states and sc-RNAseq distributions)—is sufficient to preliminarily validate the methodology and model.

That said, we certainly recognize the importance of nonlinear dimensionality reduction methods like UMAP and t-SNE in analyzing complex phenomena like tumor heterogeneity and cell state transitions. For instance, in this work, we simulated common mutations in glioma, which can be used to further explore both the internal and external heterogeneity of glioma, although this is not the primary focus of the current work. In our previous work [27], we also investigated glioma cell state transitions from a bottom-up perspective and discovered that different subtypes of glioma cells might transform into each other through a cluster of transition states, including the glioma stem cell state. In subsequent studies on these features, nonlinear dimensionality reduction algorithms like UMAP/t-SNE are indispensable.

### 3.2. Potential Clinical Applications

The methodology presented in this work serves as an efficient tool for screening therapeutic targets, allowing for the prioritization of potentially effective combinations. Subsequently, existing or new medications can be applied, thereby reducing the costs of development and accelerating clinical realization of pharmaceutical research.

If a target gene is inactivated due to mutations or deletions, activating it through chemical agonists may not be an effective therapeutic approach. Our method can indeed simulate such scenarios, and we have explored the most effective intervention combinations when encountering such cases. For example, we found that when P53 is set to 0, upregulating PTEN while downregulating Akt could reduce the tumor attractor domain to 6.47%, which is about 22.84% of the attractor size in the no intervention group (28.32%) or 13.36% of the size in the P53 mutation group (48.40%), and this result comes from a random exploration covering only 2.7% of the total combination space.

Gene editing technologies, such as CRISPR, have been applied to repair mutated genes for cancer treatment [35]. CRISPR is indeed a revolutionary gene editing technology, but before its emergence, technologies like TALEN and ZNF already existed. However, gene editing has not yet been widely applied to cure tumors. Apart from the technical limitations, such as the difficulty in precise drug delivery and off-target effects, the nonlinear nature of gene regulatory networks is also a key factor hindering their development. Similarly, studies have shown that “oncogenic mutations” are widely present even in healthy individuals [36], so why do these individuals not develop tumors? Answering such questions also requires considering the nonlinear nature of gene regulatory networks.

In previous work [27], including this study, we found that the inactivation of “tumor suppressor genes” like P53, P16, PTEN, and RB sometimes only expands the glioma attractor domain, but the barrier between non-tumor and tumor states still exists. The system still requires significant perturbations, such as further mutations leading to the complete disappearance of the tumor/non-tumor barrier, or the unavoidable accumulation of long-term small perturbations, to spontaneously transition from a non-tumor steady state to a tumor steady state. Conversely, even if all mutated genes are repaired, the transition from the tumor state to the non-tumor state (e.g., apoptosis or normal cell state) still requires overcoming the barrier. Repairing all mutated genes is a massive task for an individual, and not all mutations need to be repaired. Our approach can be used to optimize the minimum gene set required to construct the tumor/non-tumor barrier. It can also be applied after gene repair to explore the optimal intervention combinations necessary to drive the system from the tumor state to the non-tumor state.

## 4. Materials and Methods

### 4.1. Model Construction

The rationale for model construction has been detailed in prior studies [19]. In brief, based on the hypothesis of the “existence of a core network”, we first selected 10 consensus functional modules and signaling pathways that are important in glioma [33,37], followed by the selection of their well-established core nodes and interactions to construct the network. For this network, we further coarse-grained unidirectional chains of action and functionally similar nodes into a single node, thereby constructing the network described in this study (Figure 1A). The experimental basis [38,39,40,41,42,43,44,45,46,47,48,49,50,51,52,53,54,55,56,57,58,59,60,61,62,63,64,65,66,67,68,69,70,71,72,73,74,75,76,77,78,79,80,81,82,83,84,85,86,87,88,89,90,91,92,93,94,95,96,97,98,99,100,101,102,103,104,105,106,107,108,109,110,111,112,113,114] for each edge can be found in Appendix A.

As a foundation for future work, we aim to avoid parameter fitting at this stage as much as possible. First, our work assumes that the core network nodes should be equally important, and thus, we used a normalized form of the Hill function. To center the S-shaped curve as much as possible, the condition *k* ≈ 2*h* must be satisfied, where *k* is the inverse of the apparent dissociation constant and *h* is the Hill coefficient.

In previous work, we observed bifurcations in several models when h was small. For this model, significant bifurcations were observed when *h* ≤ 3 [27]. We interpret this phenomenon as the evolution of secondary interactions involving h, and this was preliminarily verified in the development model of the telencephalon anterior–posterior patterning [115]. At *h* = 3, *k* = 10, and for *h* = 4, 5, 6, …, 10, both the number and types of fixed points remained the same [27]. Therefore, in this work, the parameter set we used was *h* = 3, *k* = 10.

The functions used in this study can be found in Appendix A.

### 4.2. Dynamic Simulations

In the current work, we used Hill-type “reaction rates” in the ordinary differential equations for dynamic calculations. Furthermore, an extra layer of regulation was introduced to the set of degradation terms [31], as illustrated in the Akt regulation equation (Figure 1B). To simulate therapeutic interventions, we assigned extreme values to the relevant nodes and regulation equations: 0 for inhibition or mutation, and 1 for activation.

We first examined the non-intervention trial to decide the number of random samples. Both the 10,000 and 100,000 random samples showed a distribution of four clusters of stable states, and the distribution patterns were similar. Considering the objective of this work is to identify intervention strategies that minimize glioma stable states, 10,000 random initial values are sufficient to capture the overall distribution structure.

Our model comprises 50 variables, and we used MATLAB’s (v.R2023b) rand function to generate random numbers uniformly distributed in the interval [0,2] with 15 decimal places, resulting in a total sample size of 10^(15 × 50). For the non-intervention group, we calculated with 100,000 random initial values 10 times using MATLAB’s fsolve function. Considering the trade-off between computational power and sampling accuracy, we used 10,000 random initial values for the simulation of the intervention group.

Given the vast difference in magnitude between the total sampling space and the sample size (10^(15 × 50) vs. 10^4–10^5), simply increasing the sample size would not necessarily provide a valid solution if a bias truly exists. To mitigate potential bias arising from the numerical methods, in previous work [27], we used different ODE solving strategies—Newton’s method and Euler’s method—for parallel comparison. Across random initial values ranging from 10,000 to 10^8, both methods consistently identified the key states of interest in this study: tumor stable states and apoptotic states, with the attractor domain size remaining stable.

Based on the above reasons, we chose a total of 100,000 uniformly distributed random initial values for the non-intervention group and 10,000 for each intervention group of simulations.

### 4.3. Data Analysis

#### 4.3.1. scRNA-Seq Data Preprocessing

We obtained the data from www.gbmseq.org [32] (this dataset can also be accessed through NCBI GEO accession: GSE84465) and selected the neoplasm, OPC, OG, and AS clusters based on their t-SNE results (Figure 2B). The logarithm of the original data was taken to approximate a normal distribution. To prevent bias from unequal sample sizes, we expanded the number of data samples in each of the four clusters to similar levels, then normalized them by calculating z-scores for each gene (by the scale function in R-v.4.0.5), followed by applying Max-Min normalization through the sweep function in R.

##### 4.3.2. Theoretical–Experimental Comparisons

We employed PCA to compare theoretical simulation results with experimental data.

A random sample of 1000 results from the untreated simulation group was taken and combined with the preprocessed single-cell sequencing data to calculate the gene correlation matrix (by the cor function in R). We computed the eigenvalues of this matrix (by eigen function in R) and selected the eigenvectors corresponding to the two largest eigenvalues as standard reference vectors for comparison. All simulated results were then projected onto these standard reference vectors, along with the preprocessed scRNA-seq data, to achieve dimensionality reduction under the same criteria, then visualized with the ggplot2 package in R. This process enabled all simulated results to be evaluated on a consistent basis. Based on the comparison between the untreated simulation results and experimental data, simulated results with a principal component 1 (PC1) value less than 1 were classified as stable tumor states.

## 5. Conclusions

This study systematically explored the potential therapies with multi-target combinations, starting with constructing and validating an endogenous network model for glioblastoma. We identified variations in the effectiveness of different target intervention combinations in suppressing tumor stable states and proposed several target combinations that may lead to a drastic reform of the landscape to completely eliminate glioma stable states.

Our study reaffirms the limitations of single-target interventions in glioma at a theoretical level. Based on the simulation results, even the most effective single-target intervention could not completely eliminate tumor stable states. This aligns with clinical observations that single-target therapies achieve only limited efficacy. In contrast, dual- and triple-target interventions substantially improved the outcomes, with 1 in 280 dual-target and 8 in 500 triple-target random simulations entirely eliminating the tumor stable states. These findings highlight the superiority and potential of multi-target strategies to overcome the challenges posed by tumor heterogeneity and adaptive resistance. By identifying combinations that disrupt key regulatory pathways, the study offers a foundation as well as a working platform for developing more precise and individualized therapies.

## Figures and Tables

**Figure 1 ijms-26-03283-f001:**
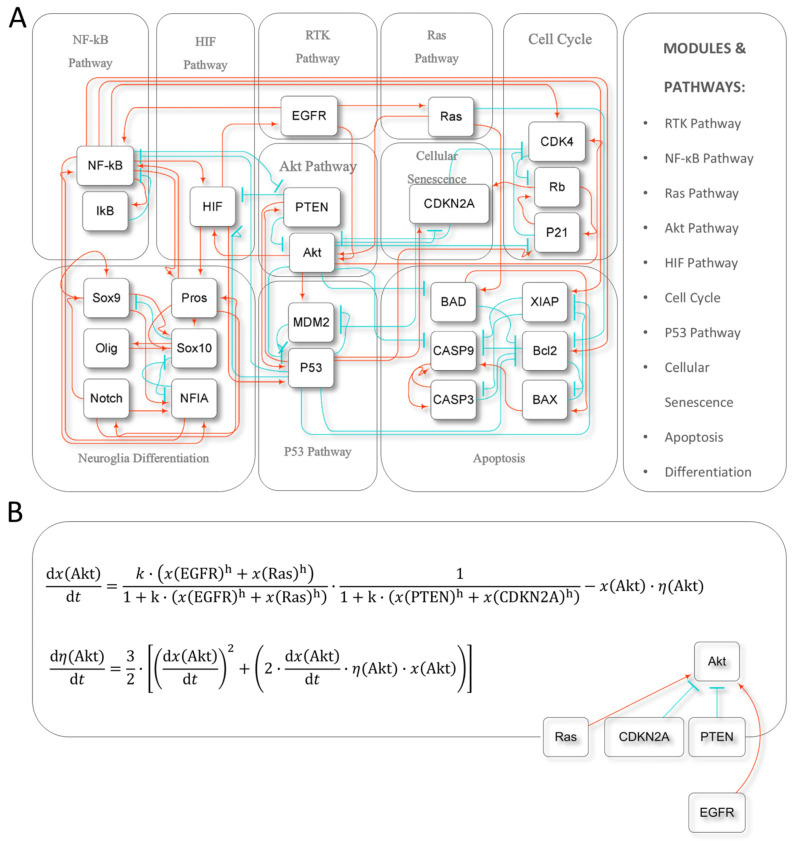
Network model and an example of the dynamics equations. (**A**) The core endogenous network for glioma comprises 25 nodes with 75 edges. A red arrow indicates an activation, while a blue T-shape sets an inhibition on the target node. (**B**) The dynamic equations for Akt are illustrated as an example. Here, *x* represents the concentration/activity of the core node, *h* is the Hill coefficient, *k* is the inverse of the apparent dissociation constant, and η denotes the degradation rate of the node.

**Figure 2 ijms-26-03283-f002:**
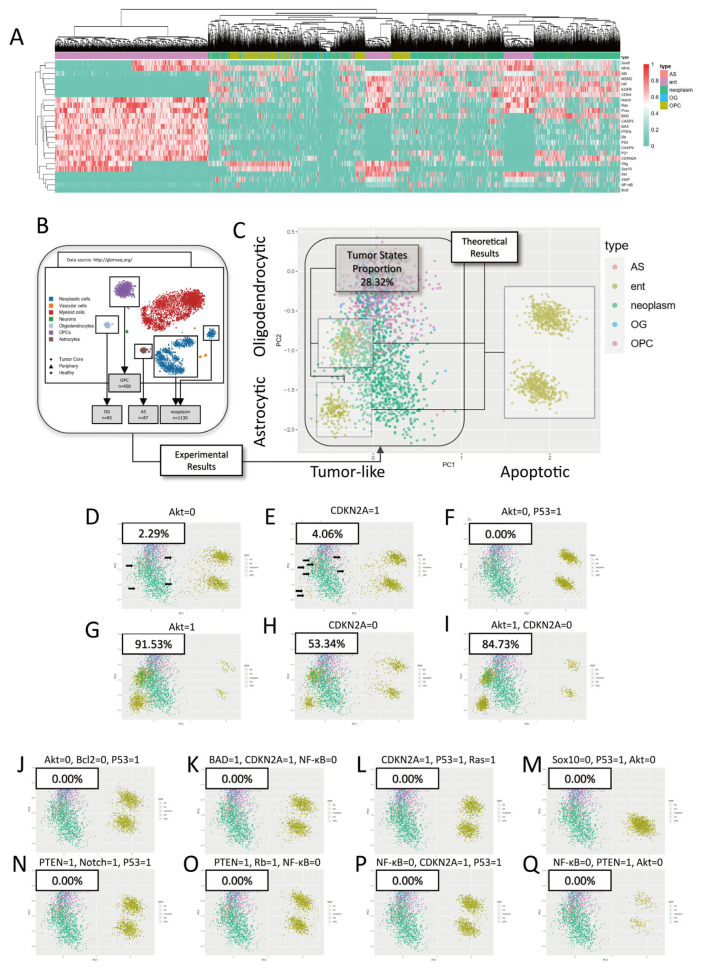
Visualization of the simulation results. (**A**) Heatmap of simulation with no intervention combined with scRNA-seq data. (**B**) scRNA-seq data selected for validation. Source: modified from www.gbmseq.org (**C**–**Q**). Distributions of modeling results combined with scRNA-seq data with principal component analysis. The title of each block represents the intervention. (**C**) is from a non-intervention simulation, while (**D**,**E**,**G**,**H**) are the results of single-target interventions. (**F**,**I**) demonstrate the results of dual-target interventions. (**J**–**Q**) illustrate the outcome of triple-target interventions. In block (**C**), the boxed area represents the tumor states obtained from theoretical calculations. In block (**C**–**Q**), the numbers indicate the proportion of tumor states among all stable states. In the (**A**,**C**–**Q**) blocks of this figure, AS represents the astrocytes, OPC stands for oligodendrocyte progenitor cells, OG stands for the oligodendrocytes, and “neoplasm” represents the tumor cells. These results came from published experimental data. The “ent” represents the results from the theoretical simulation.

**Table 1 ijms-26-03283-t001:** Proportion of tumor stable states upon single-target intervention.

Target = 1		Target = 1	
Intervention	Glioma States Proportion	Intervention	Glioma States Proportion
Akt = 0	2.29%	Akt = 1	91.53%
CDKN2A = 1	4.06%	MDM2 = 1	61.06%
P53 = 1	6.09%	CDKN2A = 0	53.34%
PTEN = 1	7.56%	Bcl2 = 1	52.61%
HIF = 0	25.15%	XIAP = 1	48.93%
BAD = 0	25.55%	P53 = 0	48.40%
Rb = 1	25.84%	NF-kB = 1	48.28%
BAX = 0	26.86%	CDK4 = 1	48.20%
IkB = 1	26.93%	PTEN = 0	46.68%
P21 = 1	27.09%	Notch = 0	44.54%
Notch = 1	27.67%	Rb = 0	42.24%
NF-kB = 0	27.95%	IkB = 0	39.81%
Ras = 0	28.08%	CASP3 = 0	38.31%
Sox9 = 0	28.11%	EGFR = 1	38.08%
Sox10 = 0	28.13%	NFIA = 1	37.88%
** No Intervention **	** 28.32% **	HIF = 1	36.66%
Olig = 1	28.43%	BAD = 1	36.44%
Bcl2 = 0	28.55%	P21 = 0	35.78%
MDM2 = 0	29.06%	Ras = 1	35.76%
BAX = 1	29.85%	NFIA = 0	35.48%
CASP9 = 0	30.26%	Sox10 = 1	35.01%
EGFR = 0	30.26%	CASP3 = 1	34.98%
Pros = 0	31.25%	Pros = 1	34.75%
XIAP = 0	31.69%	CASP9 = 1	34.60%
CDK4 = 0	33.45%	Olig = 0	34.38%
Sox9 = 1	33.60%		

The red color and bold mark the baseline of the model results.

**Table 2 ijms-26-03283-t002:** Proportion of tumor stable states in dual- and triple-target simulations.

Targets = 2		Targets = 2		Targets = 3		Targets = 3	
Intervention	Glioma States Proportion	Intervention	Glioma States Proportion	Intervention	Glioma States Proportion	Intervention	Glioma States Proportion
Akt = 0	0.00%	Akt = 1	84.74%	NF-kB = 0	0.00%	IkB = 0	100.00%
P53 = 1		CDKN2A = 0		CDKN2A = 1		Olig = 0	
				P53 = 1		Akt = 1	
P21 = 1	0.80%	Akt = 1	83.28%	Akt = 0	0.00%	Akt = 1	100.00%
Akt = 0		Ikb = 0		Bcl2 = 0		Rb = 0	
				P53 = 1		NFIA = 0	
CDKN2A = 1	1.24%	Akt = 1	79.62%	CDKN2A = 1	0.00%	IkB = 0	100.00%
PTEN = 1		CASP9 = 0		P53 = 1		P53 = 0	
				Ras = 1		P21 = 0	
Akt = 0	1.27%	Notch = 0	78.44%	PTEN = 1	0.00%	Sox10 = 0	100.00%
Bcl2 = 1		Akt = 1		Notch = 1		Bcl2 = 1	
				P53 = 1		Akt = 1	
Akt = 0	1.33%	XIAP = 0	78.27%	PTEN = 1	0.00%	Rb = 0	100.00%
BAD = 1		Akt = 1		Rb = 1		Akt = 1	
				NF-kB = 0		P21 = 1	
Akt = 0	1.34%	XIAP = 1	77.82%	Sox10 = 0	0.00%	Akt = 1	100.00%
Notch = 1		Akt = 1		P53 = 1		IkB = 0	
				Akt = 0		Notch = 0	
Akt = 0	1.34%	Akt = 1	73.64%	NF-kB = 0	0.00%	BAX = 0	100.00%
XIAP = 1		MDM2 = 0		PTEN = 1		Rb = 1	
				Akt = 0		Akt = 1	
Akt = 0	1.40%	NF-kB = 1		BAD = 1	0.00%	Akt = 1	100.00%
Sox9 = 1		CDKN2A = 0	68.83%	CDKN2A = 1		Sox10 = 1	
				NF-kB = 0		XIAP = 0	
Pros = 1	1.45%	P53 = 0		CDKN2A = 1	0.08%	NFIA = 0	100.00%
Akt = 0		PTEN = 0	55.99%	Notch = 1		Akt = 1	
				PTEN = 1		BAD = 0	
Olig = 1	1.46%	CDKN2A = 0		Akt = 0	0.09%	IkB = 0	100.00%
Akt = 0		Rb = 0	52.51%	BAX = 1		NFIA = 0	
				CDKN2A = 1		Akt = 1	
BAD = 0	1.46%	CDKN2A = 0		Ras = 1	0.12%	P21 = 0	100.00%
Akt = 0		BAX = 1	51.37%	Rb = 1		Bcl2 = 1	
				PTEN = 1		Notch = 0	
XIAP = 0	1.47%	PTEN = 0		P21 = 1	0.14%	Akt = 1	100.00%
Akt = 0		NF-kB = 1	50.21%	PTEN = 1		Rb = 0	
				Bcl2 = 0		EGFR = 0	
NFIA = 1	1.49%	HIF = 0		Olig = 1	0.17%	BAD = 1	100.00%
Akt = 0		P53 = 0	48.87%	NF-kB = 0		Pros = 0	
				CDKN2A = 1		Akt = 1	
Akt = 0	1.51%	BAX = 1		P53 = 1	0.35%	Pros = 0	100.00%
Sox10 = 1		P53 = 0	48.75%	XIAP = 1		P53 = 0	
				PTEN = 1		P21 = 0	
EGFR = 1	1.59%	Notch = 1		PTEN = 1	0.48%	CASP9 = 0	100.00%
Akt = 0		P53 = 0	48.27%	Pros = 1		XIAP = 0	
				MDM2 = 0		Akt = 1	
Olig = 0	1.68%	XIAP = 1		P53 = 1	0.60%	Olig = 1	100.00%
Akt = 0		CDKN2A = 0	47.84%	Bcl2 = 1		Akt = 1	
				PTEN = 1		BAX = 1	
CDKN2A = 1	3.23%	P53 = 0		CASP9 = 1	0.73%	Akt = 1	100.00%
NF-kB = 0		P53 = 0	47.79%	Ras = 1		Sox10 = 0	
				Akt = 0		Bcl2 = 1	
Ras = 0	3.64%	P53 = 0		EGFR = 1	0.79%	Akt = 1	100.00%
CDKN2A = 1		Pros = 1	47.71%	CDKN2A = 1		CASP9 = 1	
				BAX = 1		Notch = 0	
CDKN2A = 1	3.73%	XIAP = 0		PTEN = 1	0.96%	Akt = 1	100.00%
BAX = 0		P53 = 0	47.58%	NF-kB = 0		Notch = 0	
				Notch = 1		XIAP = 1	
CDKN2A = 1	3.80%	Olig = 0		HIF = 0	1.02%	IkB = 0	100.00%
Sox10 = 1		P53 = 0	47.58%	CASP9 = 1		BAD = 0	
				PTEN = 1		Akt = 1	
NFIA = 1	3.91%	P53 = 0		Akt = 0	1.09%	Akt = 1	100.00%
CDKN2A = 1		XIAP = 1	47.26%	CASP9 = 1		Ras = 0	
				Sox9 = 1		PTEN = 1	
EGFR = 1	4.15%	P21 = 0		CDKN2A = 1	1.17%	Akt = 1	100.00%
CDKN2A = 1		PTEN = 0	47.12%	Sox10 = 1		Rb = 0	
				Rb = 1		BAD = 1	
P53 = 1	4.96%	NFIA = 1		PTEN = 1	1.31%	Akt = 1	100.00%
MDM2 = 0		P53 = 0	46.87%	Bcl2 = 0		MDM2 = 1	
				HIF = 0		P21 = 0	
P53 = 1	5.42%	EGFR = 0		CASP9 = 1	1.46%	Akt = 1	100.00%
Ras = 0		P53 = 0	46.64%	P53 = 1		IkB = 1	
				NF-kB = 0		HIF = 0	
PTEN = 1	5.75%	Ras = 0		PTEN = 1	1.47%	Akt = 1	100.00%
MDM2 = 0		CDKN2A = 0	46.60%	Pros = 0		NFIA = 1	
				MDM2 = 0		Olig = 1	

## Data Availability

Data from the analyzed model results are provided in Appendix A. The sc-RNAseq data of glioblastoma used in this study can be accessed at www.gbmseq.org, maintained by GEPHART LAB, STANFORD NEUROSURGERY, or by NCBI GEO accession: GSE84465.

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
