# Peer review of "Exploring Multi-Target Therapeutic Strategies for Glioblastoma via Endogenous Network Modeling"

_ijms, 2025, doi:10.3390/ijms26073283_

Round 1
Reviewer 1 Report
Comments and Suggestions for Authors
This study makes meaningful contributions to understanding multi-target therapeutic strategies for glioblastoma. The combination of computational modeling with experimental validation provides insights into future therapeutic development. However, some questions should be addressed.
Major comments
Q1) The paper doesn't fully explain the rationale for including specific nodes in the endogenous network
Figure 1A shows the network structure, but doesn't detail:
Why were certain interactions included or excluded? Experimental knowledge is not sufficient. The evidence supporting each edge in the network. The relative weights or importance of different connections.
Q2) The Hill function implementation (described in section 4.2) needs more detail:
The specific form of the Hill equations used
How parameters were chosen
Sensitivity analysis of the results to parameter changes
Q3) The authors used PCA to analyze and compare their simulation results with scRNA-seq data. However, this choice raises a significant methodological concern. Single-cell RNA sequencing data is characterized by high dimensionality, sparsity, strong nonlinear relationships between genes, and complex cellular hierarchies. These characteristics make nonlinear dimensionality reduction methods like UMAP and t-SNE the standard choice in scRNA-seq analysis, as they better preserve local structure, handle nonlinear relationships, and reveal cellular heterogeneity. PCA, being a linear method, may miss important biological patterns in the data.
Specifically, for glioblastoma analysis:
- How might the choice of PCA over UMAP/t-SNE affect the validation of their model against scRNA-seq data?
- Could key aspects of tumor heterogeneity and cell state transitions be overlooked due to PCA's linear nature?
- Would using nonlinear methods reveal additional insights about the relationship between their simulated states and actual cellular states?
The authors should justify their choice of PCA or consider validating their results using nonlinear dimensionality reduction methods that are more appropriate for scRNA-seq data analysis.
Q4) The choice of 100,000 initial conditions for non-intervention and 10,000 for intervention groups needs justification
The paper should address:
- How these numbers were determined?
- Is this 10:1 ratio based on any statistical considerations? Could the different sample sizes introduce bias when comparing intervention vs non-intervention results?
- Whether the results are stable with different sample sizes.
Author Response
Q1) The paper doesn't fully explain the rationale for including specific nodes in the endogenous network. Figure 1A shows the network structure, but doesn't detail:
Q1.1) Why were certain interactions included or excluded?
Response 1.1: Thank you for your comments. In response to your suggestion, we have added further details on the rationale behind the model construction in the discussion section. See lines 273-297.
The details and rationale of this model have been thoroughly explained in previous work (Yao, et al., 2024). In brief, based on the hypothesis of the "existence of a core network," we first selected consensus functional modules and signaling pathways that are important in glioma, followed by the selection of their well-established core nodes and interactions to construct the network. For this network, we further coarse-grained unidirectional chains of action and functionally similar nodes into a single node, thereby constructing the network described in this study.
Q1.2) Experimental knowledge is not sufficient. The evidence supporting each edge in the network.
Response 1.2: In accordance with your suggestion, we have included experimental evidence for each edge in the network in the supplementary materials.
We agree with your point that the current experimental knowledge is far from sufficient to construct a detailed network, so we have built a coarse-grained core network model. Even so, this model is sufficient to generate theoretical predictions that cannot be obtained solely through statistical methods.
Q1.3) The relative weights or importance of different connections.
Response 1.3: We appreciate you raising this issue, and we have added relevant discussion in the updated discussion section. See lines 168-172.
As explained in the previous section and our earlier work, experimental knowledge is insufficient to truly determine the edge weights, which are detailed parameters. Instead of fitting data, we started from a theoretical basis, assuming that the nodes and edges within the core network are "equally important." Therefore, after normalizing the concentration/activity of the nodes, the weights of all edges are considered to be the same.
Q2) The Hill function implementation (described in section 4.2) needs more detail:
Q2.1) The specific form of the Hill equations used
Response 2.1: Thank you for your comment. We have added additional descriptions in the methodology section (see line 322) and included supplementary material that records the specific form of the equations used.
Q2.2) How parameters were chosen
Q2.3) Sensitivity analysis of the results to parameter changes
Response 2.2 & 2.3: Based on your suggestion, we have added the relevant explanation to the Materials and Methods section. See lines 307-329.
As a foundation for future work, we aim to avoid parameter fitting at this stage as much as possible. First, our work assumes that the core network nodes should be equally important, and thus, we used a normalized form of the Hill function. To center the S-shaped curve as much as possible, the condition k≈2h must be satisfied, where k is the inverse of the apparent dissociation constant and h is the Hill coefficient.
In previous work, we observed bifurcations in several models when h was small. For this model, significant bifurcations were observed when h≤3. We interpret this phenomenon as the evolution of secondary interactions involving h, and this was preliminarily verified in the development model of the telencephalon anterior-posterior patterning (Sun et al., 2024).
At h=3, k=10, and for h=4, 5, 6, …,10, both the number and types of fixed points remained the same. Additionally, studies have shown that in the evolution of Hill-form public goods games, the Hill coefficient tends to evolve to around 3 (Archetti & Scheuring, 2016). Therefore, in this work, the parameter set we used was h=3, k=10.
Q3) The authors used PCA to analyze and compare their simulation results with scRNA-seq data. However, this choice raises a significant methodological concern. Single-cell RNA sequencing data is characterized by high dimensionality, sparsity, strong nonlinear relationships between genes, and complex cellular hierarchies. These characteristics make nonlinear dimensionality reduction methods like UMAP and t-SNE the standard choice in scRNA-seq analysis, as they better preserve local structure, handle nonlinear relationships, and reveal cellular heterogeneity. PCA, being a linear method, may miss important biological patterns in the data. Specifically, for glioblastoma analysis:
Q3.1) How might the choice of PCA over UMAP/t-SNE affect the validation of their model against scRNA-seq data?
Q3.2) Could key aspects of tumor heterogeneity and cell state transitions be overlooked due to PCA's linear nature?
Q3.3) Would using nonlinear methods reveal additional insights about the relationship between their simulated states and actual cellular states?
Q3.4) The authors should justify their choice of PCA or consider validating their results using nonlinear dimensionality reduction methods that are more appropriate for scRNA-seq data analysis.
Response 3.1, 3.2, 3.3 & 3.4: Thank you for your suggestion. Based on your feedback, we have expanded our discussion of this topic in both the Results and Discussion sections of the manuscript. See lines 97-103 and 179-210.
We have tried using the t-SNE method to compare the theoretical results with single-cell data and found that t-SNE tends to amplify differences, while our study is more focused on identifying commonalities. For the steady states obtained from our current calculations, hierarchical clustering combined with biological knowledge suggests that the main distinctions are: proliferation/apoptosis and astrocytic-like/oligodendrocytic-like, and PCA is sufficient to distinguish these two main features. The interpretable and reproducible nature of PCA also makes it more suitable for observing how different interventions impact the simulated results under the same standard.
Yes, the dimensionality reduction results from PCA are far less detailed than those from UMAP/t-SNE. Therefore, from a data-driven perspective, using only PCA is likely to overlook characteristics such as tumor heterogeneity and cell state transitions.
However, in the context of this study, our method is not limited to data analysis. We adopt a bottom-up perspective, where experimental data analysis serves to validate theoretical predictions. For this purpose, we prefer using PCA, a straightforward, reproducible, and interpretable method.
That said, we certainly recognize the importance of tumor heterogeneity and cell state transitions. For instance, in this work, we simulated common mutations in glioma, which can be used to further explore both the internal and external heterogeneity of glioma, although this is not the primary focus of the current work. In our previous work (Yao, et al, 2024), we also investigated glioma cell state transitions from a bottom-up perspective and discovered that different subtypes of glioma cells might transform into each other through a cluster of transition states, including the glioma stem cell state. In subsequent studies on these features, nonlinear dimensionality reduction algorithms like UMAP/t-SNE are indispensable.
Q4) The choice of 100,000 initial conditions for non-intervention and 10,000 for intervention groups needs justification. The paper should address:
Q4.1) How these numbers were determined?
Q4.2) Is this 10:1 ratio based on any statistical considerations? Could the different sample sizes introduce bias when comparing intervention vs non-intervention results?
Q4.3) Whether the results are stable with different sample sizes.
Response 4.1, 4.2 & 4.3: Thank you for your suggestion, and we have added the relevant justification in the methods and material section. See lines 336-361.
We first examined the non-intervention trial to decide the number of random samples. Both the 10,000 and 100,000 random samples showed a distribution of four clusters of stable states, and the distribution patterns were similar. Considering the objective of this work is to identify intervention strategies that minimize glioma stable states, 10,000 random initial values are sufficient to capture the overall distribution structure.
Our model includes 50 variables, with sampling conducted on four decimal floating-point numbers uniformly distributed in the [0, 2] range. The total sampling space is on the order of 10^30. For the non-intervention group, we calculated with 100,000 random initial values for 10 times using MATLAB’s fsolve function, with a total computation time of 280 minutes. For the intervention groups, we calculated over 800 random intervention combinations. Considering the trade-off between computational power and sampling accuracy, we used 10,000 random initial values, with a total computation time of approximately 2 weeks.
Given the vast difference in magnitude between the total sampling space and the sample size, simply increasing the sample size would not necessarily provide a valid solution if bias existed. To mitigate potential bias arising from the numerical computation methods, in previous work (Yao et al., 2024), we used different ODE solving strategies—Newton's method and Euler's method—for parallel comparison. Across random initial values ranging from 10,000 to 1e9, both methods consistently identified the key states of interest in this study: tumor stable states and apoptotic states, with the attractor domain size remaining stable.
Given the vast difference in magnitude between the total sampling space and the sample size, simply increasing the sample size would not necessarily provide a valid solution if bias existed. To mitigate potential bias arising from the numerical computation methods, in previous work (Yao et al., 2024), we used different ODE solving strategies—Newton's method and Euler's method—for parallel comparison. Across random initial values ranging from 10,000 to 1e9, both methods consistently identified the key states of interest in this study: tumor stable states and apoptotic states, with the attractor domain size remaining stable.
Based on the above reasons, we chose the sample size described in this study. Once again, thank you for your constructive suggestion.
Reviewer 2 Report
Comments and Suggestions for Authors
The researchers tackle the challenges of glioblastoma treatment by exploring the application of dynamical analysis to a previously developed endogenous network. Their approach aims to uncover critical regulatory mechanisms and potential therapeutic targets, offering new insights into the complex behavior of glioblastoma cells and paving the way for more effective treatment strategies. In their practical section, the scientists discuss cancer-related research, and as part of their discoveries and contributions, the writers emphasize the need for using multi-target intervention strategies in cancer treatment. As part of their investigation, the scientists look at previously documented endogenous networks and corroborate their findings with single-cell RNA sequencing data from glioblastoma cases.
In accordance with the above information and understanding of the research framework, I propose the following changes to improve the quality of this original research paper:
1. For all authors within this specified article, it is necessary to specify and create appropriate ORCID numbers, which are currently missing and not specified.
2. When citing all the references within the article, it is necessary to have one empty space. Specifically, to be realized in the following way as in this example: “…nervous system[1–3]…” -> change to -> “…nervous system [1–3]…”
3. On line 45, within the next sentence "A novel approach based on such concepts emphasized on the causality of the..." information is missing, which specific approach is meant? Authors should be more precise and state the name of the approach itself, as well as whether it is your approach or from another author to avoid possible later confusion.
4. On line 49, for the sentence "This type of methodology has gained increasing acceptance[17] in the literature." is currently unfinished and I have the impression that something is missing, the answer to the question why? I ask the authors to complete the above sentence and expand the context. Also, in the previous part you mention "The theory has been…" and in the next sentence you mention "This type of methodology". I suggest the authors to be careful when expressing and citing methodology or theory, whether it is one or the other.
5. On line 51, for the sentence "The landscape of glioma has been prior-established based on experimental-data-driven[18] and first-principle-like analyses[19]." there is a lack of detailed information on how specifically this part is applied within the research itself and what it is trying to apply to. I suggest the authors to state what were their initial ideas and foundations, desired inventions and proposals for new ideas that are implied for first-principle analysis. It would be desirable to state which are the fundamental axioms in the given arena that you state within this paragraph.
6. On line 55, for the sentence "The plight calls for innovative methodologies in the field." the appropriate connection with the previous paragraph is missing. I ask the authors to be more precise and to explain in which specific field (medicine, IT, auto industry, aviation, etc.)?
7. The content on line 58 of the sentence "In this article, we systematically explored multi-target therapeutic strategies..." should have been listed considerably earlier, preferably within the first few sentences of the introduction. I advise that the authors improve the structure of the entire introduction section. My suggestion is as follows:
a. In the first paragraph, write background that places the subject at hand in a larger perspective and emphasizes the study's purpose.
b. Desired and used procedures, briefly detail the key procedures or treatments used, which you have previously written in your second paragraph.
c. What are the results you expect or have achieved? Outline the primary results of the paper in a few phrases.
8. Before the end of the introductory section, it is necessary to provide information about the concrete contributions made and the location in which parts of the articles they are located. It would be desirable to list the names of the methods that you managed to develop.
9. At the end of the introduction, it is necessary to state one paragraph that contains information about the organization of the article, what it specifically contains in terms of sections. For example: "The research framework is organized into several appropriate units as follows: Introduction....., Section YZ, .... conclusion..."
10. Within the paper, the section containing the Literature Review is missing. I ask the authors to look at the current situation in the field after the introduction and to create the mentioned section. Within the mentioned section, it is necessary to look at all the works that have already made certain significant contributions. The current lack of a Literature Review section undermines the validity of the proposed methods and the very contributions you propose.
11. The third section should be Materials and Methods, which is currently very far away and is located after the results. I ask the authors to make appropriate corrections and to place the Materials and Methods section after the Literature Review.
12. Section 2. Results, must contain an appropriate introductory paragraph that presents the appropriate Laboratory environment used and does it relate to medicine or some other field? Because at the moment only the sentence on the line "...making the task challenging in laboratory..." has been mentioned. I ask the authors to state precisely whether they used computers and other medical equipment. Ideally information about their features, locations, versions and other specifics that are valid for the research framework itself within this article.
13. On line 65, "...the network nodes were regulated by Hill functions..." missing a proper reference and a more detailed explanation about the starting values ​​that were considered before normalization or only normalization was taken into account.
14. On line 68, "....related nodes 68 to extreme values...." there is missing information within the text, what exactly are these values? I ask the authors to express precisely and better connect with the mathematical formula shown in Figure 1 - Part B.
15. Currently explained within the text and the displayed model does not match and does not contain all relevant information. I ask the authors to make appropriate additions within the text. My suggestion is that the sections in Figure 1, such as: NF-kB Pathway be changed to A1 - NF-kB Pathway. After that, the presented context can be better explained and connected within the text. It is also necessary to implement this part for Modules and Pathways.
16. Within 2.2. Model Validation, it is necessary to specify specific information related to the validation of the model itself and how many there will be or you expect to realize as part of the experiments
17. The quality for all figures must be significantly higher in vector format, ideally 1,200 dpi. In order to clearly identify the author's contribution and avoid copyright issues, it is necessary to state: Source: author's contribution or state the source itself if the appropriate modification has been made. In the case of modification in order to improve quality, it is necessary to state: modified in order to improve quality based on the source [X].
18. Caption for Figure 1, as well as for all other Figures, must contain a specific name. While only the description must be moved to be inside the text
19. Subsection 2.2.1. Validation with scRNA-seq Data must be explained in more detail to better present the part that refers to all the parts shown in Figure 2. Within the text there must be a clear explanation of the A block, which is currently included within the description of Figure 2. So the text also contains an explanation the same legend that is shown on the right side for Figure 2 - Block A. I ask the authors to improve the quality for the mentioned block A, because it is currently not possible to read and understand what exactly it contains, even after zooming to 1600%. I suggest to the authors the segmentation and separation of the mentioned blocks into separate figures, where it will be easier to explain the presented achievements in terms of results.
20. Within Figure 2, block B mentions a specific data source that is not explained or mentioned anywhere within the text. I ask the authors to explain the reasons and to state in what way the mentioned source: gbmseq.org was used?
21. Within Figure 2, block C contains information with a percentage of 28.32% and where certain parts are indicated. Where it remains unclear how the mentioned percentage was arrived/created at, how was the diagram itself generated? I ask the authors to add the appropriate used code that they used to generate the diagram itself, as well as to add the appropriate correlation on the diagram itself. Because the current presentation is difficult to repeat and implement by other authors in terms of validation of the realized and shown results.
22. Further on in Figure 1, the reason for the words from F-Q is not clear as to why they are shown and it is very difficult to tell the difference. I suggest that the authors try to apply the kernel density estimate (KDE) plot, so that the differences between each segment can be clearly seen
23. On line 94 "As a part of the investigation process, we then simulated the impact of common glioblastoma mutations..." appears without a clear link to the previous paragraph or section. The current sentence disrupts the flow of the reading. I am asking the author to make appropriate additions that will make it easier to understand the process that you present within the mentioned section and the very reason that the subsection itself is created. My proposal is that the current text be combined with the previous paragraph and that the current subsection be folded, where the title itself could be better explained within the text.
24. On line 98, "Such consistency with clinical/experimental observations further consolidated the model constructed for the subsequent investigation." the current formulation is not finalized and it is unclear what the model contains/looks like and how it is constructed? Information about the accuracy of the model itself or the problems you have identified is missing. I ask the author to explain more precisely information about the model, as well as what types of observations are expected, planned or desired.
25. Within Table 1, you should mark in bold the values ​​that you consider to be the most important for your conducted research
26. Section 2.3. Exploration of Therapeutic Strategies is empty and does not contain an appropriate introductory paragraph. I ask the authors to make appropriate additions
27. Within section 2.3.1. Single-Target Interventions and Sentences on Line 103, "After the validation..." lacks information about the validation process itself, which has not been implemented anywhere before. I ask the authors to implement the first validation process and to clearly define and explain it.
28. On line 105, "First, we went through all single-target interventions and found that the downregulation of Akt achieved the best therapeutic effect, reducing tumor states from 28.32% to 2.29%...." the corresponding information is missing by which method stated claims and provided information? I ask the authors to precisely define the steps or to show the process by which 2.29% was reached through appropriate algorithmic steps.
29. The content offered in this section (2.3.1. Single-Target Interventions) is missing from the currently published abstract, which calls into question the legitimacy and quality of the scientific article's presentation. I ask the authors to improve the abstract and link the complete work included therein, which is currently not the case. There are numerous flaws in both the abstract and the substance offered as whole in the paper.
30. Section 3. Discussion could be better written, to contain a concrete discussion of the achieved results, limits and remaining open questions. Currently, on line 163 "This study offers potential clinical applications. The methodology presented in this work serves as an efficient tool for screening the therapeutic targets,..." there is a lack of information in what way and in what specific clinical applications? The second part that is problematic is related to the indication of methodology and tools, which is not presented in the mentioned way anywhere within the research. I ask the authors to precisely define what exactly it is about and the exact part it refers to.
31. Section 4. Materials and Methods, as previously stated must have been written much earlier. Within the mentioned section, there is a lack of information about the specific research method within the introductory paragraph, as well as about the application of numerical or non-numerical data. Given that it was research in medicine, it is necessary to state how the analyzed data was collected, under which license and whether the appropriate processes of protection and anonymization of the patient data were implemented.
32. On line 177, "However, no single-target intervention could entirely eliminate tumor stable states. These results would be further discussed later." the presented sentences seem incomplete and do not contain the expected information related to the creation of the model. I ask the authors to make the appropriate changes and additions, with which only the process and the part that you mention will be discussed later can be more clearly understood. It would be desirable to state within which section the mentioned results are presented.
33. On line 189, "We obtained the data from gbmseq.org[23]..." you must specify the license under which the data is published, as well as information about the consent itself if it is private data. It must also be stated within the "Data Availability Statement:" that the mentioned dataset is used. Current information in "Data Availability Statement: All data included in this study are provided in supplementary table." it is not enough and must be supplemented.
34. Within the section "4.3.1. scRNA-seq data Preprocessing" is missing appropriate information related to scRNA-seq data and how it is planned to be used? The current section is not well written and must be expanded with appropriate information that accurately illustrates the realization processes through text or visual form. I ask the authors to implement the appropriate changes.
35. On line 196, "We employed principal component analysis (PCA)" the appropriate reference and reason for the choice is missing
36. On line 199, "We computed the eigenvalues ​​of this matrix and selected the eigenvectors corresponding to the two largest eigenvalues ​​as standard..." information about the hardware used for the calculation is missing. Specifically, CPU, GPU, RAM, type of hardware architecture (ARM, x64, x86, etc.) or specifications of the medical device model, year of manufacture and its manufacturer. I ask the authors to provide precise information related to the laboratory environment so that there could be validity of all presented results, as well as the possibility of repetition by other authors
37. The conclusion section is missing within the paper
38. Before the literature, the corresponding section showing the list of used abbreviations is missing
39. The number of references within this paper must be significantly higher, considering the field of research. At the moment, there are no corresponding relevant articles that represent the starting points and the background.
40. The level of English must be substantially higher in terms of expressing precision; some sentences are incomplete and require many readings to comprehend. The sentences in the various sections I've included in the comments lack an appropriate beginning and end. Others contain newly added sentences with no clear context, confusing the reader greatly.
The currently provided research includes numerous flaws in terms of structure and presentation of the obtained scientific results. Many areas appear incomplete and do not meet the required standard of quality. Certain areas are chaotic, making it impossible to confirm the statistical data. The main issue of the current version of this article is its confusing methodology, expression in English with fragmented phrases. Where the offered abstract does not correspond to the content of the work.
Comments on the Quality of English Language
The level of English must be substantially higher in terms of expressing precision; some sentences are incomplete and require many readings to comprehend. The sentences in the various sections I've included in the comments lack an appropriate beginning and end. Others contain newly added sentences with no clear context, confusing the reader greatly.
Author Response
Q1. For all authors within this specified article, it is necessary to specify and create appropriate ORCID numbers, which are currently missing and not specified.
Response1: Thank you for your suggestion. We have provided the available ORCIDs, see lines 4-5.
Q2. When citing all the references within the article, it is necessary to have one empty space. Specifically, to be realized in the following way as in this example: “…nervous system[1–3]…” -> change to -> “…nervous system [1–3]…”
Response2: Thank you for your suggestion. We have revised the citation format throughout the manuscript. See lines 29, 30, 32, 34, 36, 40, 43, 51, 55, 59, 64, 77, 102, 120, 143, 207, 229, 234, 237, 305, 317, 320, 321, 334, 353, 364.
Q3. On line 45, within the next sentence "A novel approach based on such concepts emphasized on the causality of the..." information is missing, which specific approach is meant? Authors should be more precise and state the name of the approach itself, as well as whether it is your approach or from another author to avoid possible later confusion.
Response3: Thank you for your suggestion. We have made corresponding revisions to the text, see line 48.
Q4. On line 49, for the sentence "This type of methodology has gained increasing acceptance[17] in the literature." is currently unfinished and I have the impression that something is missing, the answer to the question why? I ask the authors to complete the above sentence and expand the context. Also, in the previous part you mention "The theory has been…" and in the next sentence you mention "This type of methodology". I suggest the authors to be careful when expressing and citing methodology or theory, whether it is one or the other.
Response4: Thank you for your suggestion. We have revised the relevant content, see lines 42-44, 51.
Q5. On line 51, for the sentence "The landscape of glioma has been prior-established based on experimental-data-driven[18] and first-principle-like analyses[19]." there is a lack of detailed information on how specifically this part is applied within the research itself and what it is trying to apply to. I suggest the authors to state what were their initial ideas and foundations, desired inventions and proposals for new ideas that are implied for first-principle analysis. It would be desirable to state which are the fundamental axioms in the given arena that you state within this paragraph.
Response5: Thank you for your suggestion. We have revised the text according to your advice, see lines 54-56.
Q6. On line 55, for the sentence "The plight calls for innovative methodologies in the field." the appropriate connection with the previous paragraph is missing. I ask the authors to be more precise and to explain in which specific field (medicine, IT, auto industry, aviation, etc.)?
Response 6: Thank you for your suggestion. We have revised the text accordingly, see lines 60-61.
Q7. The content on line 58 of the sentence "In this article, we systematically explored multi-target therapeutic strategies..." should have been listed considerably earlier, preferably within the first few sentences of the introduction. I advise that the authors improve the structure of the entire introduction section. My suggestion is as follows:
- In the first paragraph, write background that places the subject at hand in a larger perspective and emphasizes the study's purpose.
- Desired and used procedures, briefly detail the key procedures or treatments used, which you have previously written in your second paragraph.
- What are the results you expect or have achieved? Outline the primary results of the paper in a few phrases.
Response7: Thank you for your systematic suggestions. We have revised the introduction section according to your advice, see lines 42-66.
Q8. Before the end of the introductory section, it is necessary to provide information about the concrete contributions made and the location in which parts of the articles they are located. It would be desirable to list the names of the methods that you managed to develop.
Response8: Thank you for your suggestion. We have added relevant content to the introduction section, see lines 69-73.
Q9. At the end of the introduction, it is necessary to state one paragraph that contains information about the organization of the article, what it specifically contains in terms of sections. For example: "The research framework is organized into several appropriate units as follows: Introduction....., Section YZ, .... conclusion..."
Response9: Thank you for your suggestion. We have added relevant content to the introduction section, see lines 67-69.
Q10. Within the paper, the section containing the Literature Review is missing. I ask the authors to look at the current situation in the field after the introduction and to create the mentioned section. Within the mentioned section, it is necessary to look at all the works that have already made certain significant contributions. The current lack of a Literature Review section undermines the validity of the proposed methods and the very contributions you propose.
Response10: We appreciate the reviewer's comment. However, our theoretical model is highly novel and builds upon years of foundational work within our laboratory. To the best of our knowledge, no other theoretical studies directly related to our approach have been published to date. For experimental studies relevant to our work, we have already provided a comprehensive set of references, which other reviewers have acknowledged as sufficient in the introduction section. Therefore, we believe the current references adequately support the context and novelty of our theoretical framework.
Q11. The third section should be Materials and Methods, which is currently very far away and is located after the results. I ask the authors to make appropriate corrections and to place the Materials and Methods section after the Literature Review.
Response11: We thank the reviewer for their suggestion regarding the placement of the Materials and Methods section. However, we have chosen to position this section after the Results to maintain the readability and logical flow of the manuscript, as it contains extensive technical details that may disrupt the narrative if placed earlier. This structure is also consistent with several recently published articles in the journal (e.g., doi.org/10.3390/ijms26052011, Jefcoate, Larsen, et al., 2025). We believe the current organization best serves the clarity of our work.
Q12. Section 2. Results, must contain an appropriate introductory paragraph that presents the appropriate Laboratory environment used and does it relate to medicine or some other field? Because at the moment only the sentence on the line "...making the task challenging in laboratory..." has been mentioned. I ask the authors to state precisely whether they used computers and other medical equipment. Ideally information about their features, locations, versions and other specifics that are valid for the research framework itself within this article.
Response12: Thank you for your suggestion. This is a theoretical work. MATLAB was used for model simulation. Default R was used for data analysis and the R package ggplot2 was used for visualization. According to your suggestion, we have revised the text, see lines 59, 290, 297, 312, 313, 316.
Q13. On line 65, "...the network nodes were regulated by Hill functions..." missing a proper reference and a more detailed explanation about the starting values ​​that were considered before normalization or only normalization was taken into account.
Response13: Thank you for your suggestion. We have elaborated on the relevant description, see line 81.
Q14. On line 68, "....related nodes 68 to extreme values...." there is missing information within the text, what exactly are these values? I ask the authors to express precisely and better connect with the mathematical formula shown in Figure 1 - Part B.
Response14: Thank you for your suggestion. We have revised the relevant description, see lines 84-85.
Q15. Currently explained within the text and the displayed model does not match and does not contain all relevant information. I ask the authors to make appropriate additions within the text. My suggestion is that the sections in Figure 1, such as: NF-kB Pathway be changed to A1 - NF-kB Pathway. After that, the presented context can be better explained and connected within the text. It is also necessary to implement this part for Modules and Pathways.
Response15: Thank you for your suggestion. We have added details of the model in the supplementary table.
Q16. Within 2.2. Model Validation, it is necessary to specify specific information related to the validation of the model itself and how many there will be or you expect to realize as part of the experiments
Response16: Thank you for your suggestion. We have revised the relevant description, see lines 97-103, 116.
Q17. The quality for all figures must be significantly higher in vector format, ideally 1,200 dpi. In order to clearly identify the author's contribution and avoid copyright issues, it is necessary to state: Source: author's contribution or state the source itself if the appropriate modification has been made. In the case of modification in order to improve quality, it is necessary to state: modified in order to improve quality based on the source [X].
Response17: Thank you for your suggestion. Following your advice, we have re-uploaded higher-resolution images and indicated the image sources, see line 108.
Q18. Caption for Figure 1, as well as for all other Figures, must contain a specific name. While only the description must be moved to be inside the text
Response18: Thank you for your suggestion. We have revised the figure captions, see lines 87, 106.
Q19. Subsection 2.2.1. Validation with scRNA-seq Data must be explained in more detail to better present the part that refers to all the parts shown in Figure 2. Within the text there must be a clear explanation of the A block, which is currently included within the description of Figure 2. So the text also contains an explanation the same legend that is shown on the right side for Figure 2 - Block A. I ask the authors to improve the quality for the mentioned block A, because it is currently not possible to read and understand what exactly it contains, even after zooming to 1600%. I suggest to the authors the segmentation and separation of the mentioned blocks into separate figures, where it will be easier to explain the presented achievements in terms of results.
Response19: Thank you for your suggestion. We have updated the figure and added relevant descriptions, see lines 97-100.
Q20. Within Figure 2, block B mentions a specific data source that is not explained or mentioned anywhere within the text. I ask the authors to explain the reasons and to state in what way the mentioned source: gbmseq.org was used?
Response20: Thank you for your question. We have indicated the data source in Figure 2, block B, and cited the data source in the revised manuscript on lines 102, 108. The usage of the data is explained in sections 4.3.1 and 4.3.2.
Q21. Within Figure 2, block C contains information with a percentage of 28.32% and where certain parts are indicated. Where it remains unclear how the mentioned percentage was arrived/created at, how was the diagram itself generated? I ask the authors to add the appropriate used code that they used to generate the diagram itself, as well as to add the appropriate correlation on the diagram itself. Because the current presentation is difficult to repeat and implement by other authors in terms of validation of the realized and shown results.
Response21: Thank you for your suggestion. We have added an explanation on lines 112-114 and included the functions used for data analysis and visualization in section 4.3.
Q22. Further on in Figure 1, the reason for the words from F-Q is not clear as to why they are shown and it is very difficult to tell the difference. I suggest that the authors try to apply the kernel density estimate (KDE) plot, so that the differences between each segment can be clearly seen
Response22: Thank you for your suggestion. We have added explanatory text on lines 109-110, 146-147, 156-157. Given the number of groups and significant overlaps, it’s hard to enhance visualization by 2-D KDE plot.
Q23. On line 94 "As a part of the investigation process, we then simulated the impact of common glioblastoma mutations..." appears without a clear link to the previous paragraph or section. The current sentence disrupts the flow of the reading. I am asking the author to make appropriate additions that will make it easier to understand the process that you present within the mentioned section and the very reason that the subsection itself is created. My proposal is that the current text be combined with the previous paragraph and that the current subsection be folded, where the title itself could be better explained within the text.
Response 23: Thank you for your suggestion. We have revised the manuscript, see line 119.
Q24. On line 98, "Such consistency with clinical/experimental observations further consolidated the model constructed for the subsequent investigation." the current formulation is not finalized and it is unclear what the model contains/looks like and how it is constructed? Information about the accuracy of the model itself or the problems you have identified is missing. I ask the author to explain more precisely information about the model, as well as what types of observations are expected, planned or desired.
Response 24: Thank you for your suggestion. We have added relevant descriptions, see lines 240-265.
Q25. Within Table 1, you should mark in bold the values ​​that you consider to be the most important for your conducted research
Response 25: Thank you for your suggestion. We have emphasized the important value.
Q26. Section 2.3. Exploration of Therapeutic Strategies is empty and does not contain an appropriate introductory paragraph. I ask the authors to make appropriate additions
Response 26: Section 2.3 serves as the parent heading for sections 2.3.1 and 2.3.2, with specific content detailed in those subsections.
Q27. Within section 2.3.1. Single-Target Interventions and Sentences on Line 103, "After the validation..." lacks information about the validation process itself, which has not been implemented anywhere before. I ask the authors to implement the first validation process and to clearly define and explain it.
Response 27: Validation refers to the comparison between theory and experimental data/knowledge, as described in section 2.2.
Q28. On line 105, "First, we went through all single-target interventions and found that the downregulation of Akt achieved the best therapeutic effect, reducing tumor states from 28.32% to 2.29%...." the corresponding information is missing by which method stated claims and provided information? I ask the authors to precisely define the steps or to show the process by which 2.29% was reached through appropriate algorithmic steps.
Response 28: Thank you for this suggestion. The definition of this value is briefly described in section 2.1 when first mentioned, with detailed provided in section 4.3 in the revised manuscript. See line 102 and 372-383
Q29. The content offered in this section (2.3.1. Single-Target Interventions) is missing from the currently published abstract, which calls into question the legitimacy and quality of the scientific article's presentation. I ask the authors to improve the abstract and link the complete work included therein, which is currently not the case. There are numerous flaws in both the abstract and the substance offered as whole in the paper.
Response 29: Thank you for your suggestion. We have revised the abstract. See lines 14, 20-22.
Q30. Section 3. Discussion could be better written, to contain a concrete discussion of the achieved results, limits and remaining open questions. Currently, on line 163 "This study offers potential clinical applications. The methodology presented in this work serves as an efficient tool for screening the therapeutic targets,..." there is a lack of information in what way and in what specific clinical applications? The second part that is problematic is related to the indication of methodology and tools, which is not presented in the mentioned way anywhere within the research. I ask the authors to precisely define what exactly it is about and the exact part it refers to.
Response 30: Thank you for your suggestion. Based on your and other reviewers' feedback, we have extensively revised the discussion section. See lines 162-265.
Q31. Section 4. Materials and Methods, as previously stated must have been written much earlier. Within the mentioned section, there is a lack of information about the specific research method within the introductory paragraph, as well as about the application of numerical or non-numerical data. Given that it was research in medicine, it is necessary to state how the analyzed data was collected, under which license and whether the appropriate processes of protection and anonymization of the patient data were implemented.
Response 31: Thank you for your suggestion. In line with your and other reviewers' advice, we have provided a more detailed description. See lines 305-383.
Q32. On line 177, "However, no single-target intervention could entirely eliminate tumor stable states. These results would be further discussed later." the presented sentences seem incomplete and do not contain the expected information related to the creation of the model. I ask the authors to make the appropriate changes and additions, with which only the process and the part that you mention will be discussed later can be more clearly understood. It would be desirable to state within which section the mentioned results are presented.
Response 32: Thank you for pointing this out. We have revised the relevant description. See lines 305-329.
Q33. On line 189, "We obtained the data from gbmseq.org[23]..." you must specify the license under which the data is published, as well as information about the consent itself if it is private data. It must also be stated within the "Data Availability Statement:" that the mentioned dataset is used. Current information in "Data Availability Statement: All data included in this study are provided in supplementary table." it is not enough and must be supplemented.
Response 33: Thank you for your suggestion. We have revised the relevant description. See lines 364-365, 394-397.
Q34. Within the section "4.3.1. scRNA-seq data Preprocessing" is missing appropriate information related to scRNA-seq data and how it is planned to be used? The current section is not well written and must be expanded with appropriate information that accurately illustrates the realization processes through text or visual form. I ask the authors to implement the appropriate changes.
Response 34: Thank you for your suggestion. We have added relevant descriptions. See lines 364-365.
Q35. On line 196, "We employed principal component analysis (PCA)" the appropriate reference and reason for the choice is missing
Response 35: Thank you for your suggestion. In line with your and other reviewers' advice, we have added justification for the method selection. See lines 180-211.
Q36. On line 199, "We computed the eigenvalues ​​of this matrix and selected the eigenvectors corresponding to the two largest eigenvalues ​​as standard..." information about the hardware used for the calculation is missing. Specifically, CPU, GPU, RAM, type of hardware architecture (ARM, x64, x86, etc.) or specifications of the medical device model, year of manufacture and its manufacturer. I ask the authors to provide precise information related to the laboratory environment so that there could be validity of all presented results, as well as the possibility of repetition by other authors
Response 36: Thank you for your suggestion. We have elaborated on the relevant description. See lines 376-377.
Q37. The conclusion section is missing within the paper
Response 37: Thank you for your suggestion. We have separated the conclusion section from the discussion. See lines 266-302.
Q38. Before the literature, the corresponding section showing the list of used abbreviations is missing
Response 38: The use of abbreviations has been minimized in this work, an additional abbreviation list is currently unnecessary.
Q39. The number of references within this paper must be significantly higher, considering the field of research. At the moment, there are no corresponding relevant articles that represent the starting points and the background.
Response 39: The theoretical and methodological foundations of this work are based on prior research from our laboratory. As stated in Response 10, we have cited the foundational theoretical work and a sufficient number of directly related experimental studies/reviews within our knowledge.
Q40. The level of English must be substantially higher in terms of expressing precision; some sentences are incomplete and require many readings to comprehend. The sentences in the various sections I've included in the comments lack an appropriate beginning and end. Others contain newly added sentences with no clear context, confusing the reader greatly.
Response 40: Thank you for your tireless assistance! We hope that with your help, the readability of this manuscript can be significantly improved.
Comment: The currently provided research includes numerous flaws in terms of structure and presentation of the obtained scientific results. Many areas appear incomplete and do not meet the required standard of quality. Certain areas are chaotic, making it impossible to confirm the statistical data. The main issue of the current version of this article is its confusing methodology, expression in English with fragmented phrases. Where the offered abstract does not correspond to the content of the work.
Response: We have undergone heavy revisions according the suggestions raised in your comments, which we believe to a point that can change the overall perception on the article.
Many of the confusions regarding the research stem from the novelty of this manuscript. It is inevitable that any new methodology will give rise to similar issues.
Nevertheless, the uniqueness and forward-looking nature of this work, along with its provision of a seemingly promising and effective tool for multi-target therapy research, justify its publication after addressing the critiques.
Reviewer 3 Report
Comments and Suggestions for Authors
This is a concise and clearly written report on a numerical simulation study related to cancer pathways, specifically those associated with glioblastoma. The title and the figures are appropriate, and overall length of the paper matches its substance well.
The study applied a prior constructed pathway model for glioblastoma, and the references cited for this and other endogenous network studies show that these authors have made a substantial investment to research in this context.
While the paper should, in my opinion, be accepted with only a minor revision, I want to point out some items to improve.
When Figure 2 is first discussed, for background explanation the readers should be referred to Materials and Methods section, specifically to the last paragraph of that section. Otherwise the methods used to call what is a tumor stable state in the simulations will remain unclear -- until the end where the truth is revealed. The authors might even consider giving the whole (short) explanation early on, in the context of Figure 2.
I believe that Figure 1 makes it amply clear to all readers that the dynamics of interactions in these (simplified) pathways will not be available for quite sometime into the future, and so the parameter choices for adopted dynamic models must be done by fiat choice -- as has been done. To extract statistically treatable datasets from the simulations, the authors have done random assignments of initial states. I have absolutely no problem with this aspect of the study. So this is not a request for improvement, just a comment that the authors can take as a compliment.
I would like the authors to discuss (or refute) this following view on the results. Lines 31 and 32 show clearly that most treatments are inhibitors to silence a pathway, not to activate it. Often cancer silences a pathway, or turns it off, by a mutation or a deletion, and then activating such gene will not succeed. I think this matches using inhibitors only in treatments. In contrast, in Table 2 the most effective interventions often assume that p53 or CDKN2A or something else is activated by the treatment (to value 1 shown in the table). Please add a short discussion to the paper either acknowledging this as a relevent viewpoint, or explaining a refutation to this viewpoint.
Author Response
Comments 1: When Figure 2 is first discussed, for background explanation the readers should be referred to Materials and Methods section, specifically to the last paragraph of that section. Otherwise the methods used to call what is a tumor stable state in the simulations will remain unclear -- until the end where the truth is revealed. The authors might even consider giving the whole (short) explanation early on, in the context of Figure 2.
Response 1: Thank you for your suggestion. Based on your feedback, we have provided a clearer explanation of the relevant background in the Results sections. See lines 77-80.
Comments 2: I believe that Figure 1 makes it amply clear to all readers that the dynamics of interactions in these (simplified) pathways will not be available for quite sometime into the future, and so the parameter choices for adopted dynamic models must be done by fiat choice -- as has been done. To extract statistically treatable datasets from the simulations, the authors have done random assignments of initial states. I have absolutely no problem with this aspect of the study. So this is not a request for improvement, just a comment that the authors can take as a compliment.
Response 2: Thank you, it is our pleasure to receive such feedback.
Comments 3: I would like the authors to discuss (or refute) this following view on the results. Lines 31 and 32 show clearly that most treatments are inhibitors to silence a pathway, not to activate it. Often cancer silences a pathway, or turns it off, by a mutation or a deletion, and then activating such gene will not succeed. I think this matches using inhibitors only in treatments. In contrast, in Table 2 the most effective interventions often assume that p53 or CDKN2A or something else is activated by the treatment (to value 1 shown in the table). Please add a short discussion to the paper either acknowledging this as a relevent viewpoint, or explaining a refutation to this viewpoint.
Response 3: Thank you for your constructive suggestion. Benefiting from your suggestions, we have added relevant discussions in the manuscript. See lines 219-249.
We agree with your point that if a target gene is inactivated due to mutations or deletions, activating it through chemical agonists may not be an effective therapeutic approach. Our method can indeed simulate such scenarios, and we have explored the most effective intervention combinations when encountering such cases. For example, we found that when P53 is set to 0, upregulating PTEN while downregulating Akt could reduce the tumor attractor domain to 6.47%, which is about one-fifth of the attractor size under no intervention, and this result comes from a random exploration only covering 2.7% of the total combination space.
Further expanding the discussion based on your clinical observation, we note that some researchers are exploring gene editing technologies, such as CRISPR, to repair mutated genes for cancer treatment (Tang & Shrager, 2016). CRISPR is indeed a revolutionary gene editing technology, but before its emergence, technologies like TALEN and ZNF already existed. However, gene editing has not yet been widely applied to cure tumors. Our view is that, apart from the technical limitations, such as the difficulty in precise drug delivery and off-target effects, the nonlinear nature of gene regulatory networks is also a key factor hindering its development. Similarly, studies have shown that so-called "oncogenic mutations" are widely present even in healthy individuals, so why don't these individuals develop tumors? Answering such questions also requires considering the nonlinear nature of gene regulatory networks.
In previous works (Yao, et al., 2024), including this one, we found that the inactivation of “tumor suppressor genes” like P53, P16, PTEN, and RB sometimes only expands the glioma attractor domain, but the barrier between non-tumor and tumor states still exists. The system still requires significant perturbations, such as further mutations leading to the complete disappearance of the tumor/non-tumor barrier, or the unavoidable accumulation of long-term small perturbations, to spontaneously transition from a non-tumor steady state to a tumor steady state (Ao, et al., 2008, Yuan, et al., 2017). Conversely, even if all mutated genes are repaired, the transition from the tumor state to the non-tumor state (e.g., apoptosis or normal cell state) still requires overcoming the barrier. Repairing all mutated genes is a massive task for an individual, and not all mutations need to be repaired. Our approach can be used to optimize the minimum gene set required to construct the tumor/non-tumor barrier. It can also be applied after gene repair to explore the optimal intervention combinations necessary to drive the system from the tumor state to the non-tumor state.
We plan to explore optimized therapeutic strategies for glioblastoma considering intra- and inter-individual heterogeneity in the next steps, and your suggestions have given us confidence. Thank you again for your constructive comments.
Round 2
Reviewer 2 Report
Comments and Suggestions for Authors
Based on the newly submitted version of the document, the authors have only partially applied the suggested comments, and earlier remarks remain unresolved. Therefore, I propose the following recommendations:
- Given the nature of the work (article), it is vital to clearly describe the specific contributions made in the introduction and where they are placed, which is now lacking.
- Within the section Introduction and the part concerning the organization, it is necessary to indicate the exact sections in which the mentioned information is found together with the appropriate references. So that it will be realized in the following way: Section X... I ask the authors to once again read the previous review related to this part, which concerns the organization of the article.
- The current version has a problematic organization and does not contain a literature review that looks at the state of the field. I ask the authors to read the previous review and apply the previous comments.
- Regarding section 4. Materials and Methods on line 304, I must insist that it be moved and listed after the introduction or literature review. Also, I ask the authors to take another look at the instructions for authors.
- The quality of the figure has not improved, and the flaws that were mentioned in my previous review are still present. The text is very difficult to read, especially on Figure 2. Also, there is a lack of detailed text that explains the content. I must insist that the authors develop each of the displayed blocks to improve quality and readability.
- Many sections of the article remain empty and lack crucial information. Consider one of the several "2.3. Exploration of Therapeutic Strategies.". I request that the authors write suitable content in all sections that are now blank.
- Within the current version, a corresponding more detailed conclusion is missing, which I see has been deleted in this version.
- The biggest disadvantage is the lack of acceptable references, which puts into doubt the overall validity of the research and provided results, discussion, and conclusions. As I stated in my previous review, for this type of research, there must be a large number of references that are valid for this area and the research carried out.
Author Response
Comments 1: Given the nature of the work (article), it is vital to clearly describe the specific contributions made in the introduction and where they are placed, which is now lacking.
Comments 2: Within the section Introduction and the part concerning the organization, it is necessary to indicate the exact sections in which the mentioned information is found together with the appropriate references. So that it will be realized in the following way: Section X... I ask the authors to once again read the previous review related to this part, which concerns the organization of the article.
Response 1 & 2: We appreciate your suggestion and have revised the Introduction section to explicitly outline the key contributions of this study. The specific additions are detailed in lines 75–77, including the integration of an endogenous network model for glioma which is now linked to Section 3.2 supported by references [32,33] and a computational framework for multi-target therapeutic screening which is now linked to Section 3.3.
Comments 3: The current version has a problematic organization and does not contain a literature review that looks at the state of the field. I ask the authors to read the previous review and apply the previous comments.
Response 3: We have expanded the Introduction to incorporate a comprehensive literature review addressing recent advances in multi-target therapies (lines 41–44) and limitations of correlation-based strategy (lines 61–65).
Comments 4: Regarding section 4. Materials and Methods on line 304, I must insist that it be moved and listed after the introduction or literature review. Also, I ask the authors to take another look at the instructions for authors.
Response 4: Following your recommendation, we have relocated the Materials and Methods section to immediately follow the Introduction (now Section 2, lines 79–150).
Comments 5: The quality of the figure has not improved, and the flaws that were mentioned in my previous review are still present. The text is very difficult to read, especially on Figure 2. Also, there is a lack of detailed text that explains the content. I must insist that the authors develop each of the displayed blocks to improve quality and readability.
Response 5: We have redesigned Figure 2 to enhance clarity:
- Text legibility was improved by increasing font size and contrast.
- Resolution was enhanced to 1200 dpi.
- Detailed annotations were added on panel B and C as the example to improve the understandability and readability of panels D-Q.
Comments 6: Many sections of the article remain empty and lack crucial information. Consider one of the several "2.3. Exploration of Therapeutic Strategies.". I request that the authors write suitable content in all sections that are now blank.
Response 6: We have addressed formatting inconsistencies in subheadings to insure the correct presentation of each section. For instance, the subheadings 3.3.1 Single-Target Interventions and 3.3.2 Multi-Target Interventions have been properly placed under the main heading 3.3 Exploration of Therapeutic Strategies, in accordance with the IJMS template requirements.
Comments 7: Within the current version, a corresponding more detailed conclusion is missing, which I see has been deleted in this version.
Response 7: We have restored the conclusion section, which now includes two paragraphs summarizing this study’s implications. See lines 322-337.
Comments 8: The biggest disadvantage is the lack of acceptable references, which puts into doubt the overall validity of the research and provided results, discussion, and conclusions. As I stated in my previous review, for this type of research, there must be a large number of references that are valid for this area and the research carried out.
Response 8: We have substantially expanded the reference list, incorporating additional citations from related works and integrating supplementary experimental basis for model construction into the main text. See lines 41-44, 61-65, 83-88, and reference list.
Round 3
Reviewer 2 Report
Comments and Suggestions for Authors
The article has been greatly enhanced from the last submitted version in terms of different features, as well as the presentation of the study technique itself. The authors have successfully implemented all of the necessary requirements.
There is one more small request that is now not accessible within the paper: the authors' ORCID numbers, as the paper was written in Word.
I believe that the article can be taken into further processing.
Author Response
Thank you for your positive feedback and for acknowledging the improvements made to the manuscript. We are grateful for your time and constructive evaluation.
Comments 1: There is one more small request that is now not accessible within the paper: the authors' ORCID numbers, as the paper was written in Word.
Response 1: Regarding your request to include ORCID numbers: We have now added the ORCID identifiers for all authors to the title page of the revised Word document.